# Developmental Parallels Between the Human Organs of Zuckerkandl and Adrenal Medulla

**DOI:** 10.3390/life15081214

**Published:** 2025-07-31

**Authors:** Ekaterina Otlyga, Dmitry Otlyga, Olga Junemann, Yuliya Krivova, Alexandra Proshchina, Anastasia Kharlamova, Victoria I. Gulimova, Gleb Sonin, Sergey Saveliev

**Affiliations:** Avtsyn Research Institute of Human Morphology of Federal State Budgetary Scientific Institution “Petrovsky National Research Centre of Surgery”, 117418 Moscow, Russia; otlyga@bk.ru (D.O.); junemann@outlook.com (O.J.); homulkina@rambler.ru (Y.K.); proshchina@yandex.ru (A.P.); grossulyar@gmail.com (A.K.); gulimova@yandex.ru (V.I.G.); glebs0nin@yandex.ru (G.S.); embrains@mail.ru (S.S.)

**Keywords:** adrenal medulla, organs of Zuckerkandl, chromaffin tissue, paraganglia, immunohistochemistry

## Abstract

The adrenal medulla and organs of Zuckerkandl consist of chromaffin cells that produce, store, and secrete catecholamines. In humans, the adrenal medulla is known to function throughout postnatal life, while the organs of Zuckerkandl degenerate by 2–3 years of postnatal life. Although the history of investigation of chromaffin cells goes back more than a century, little is known about the reciprocal organogenesis of the adrenal glands and organs of Zuckerkandl during human fetal development. In the current study, we compared these two organs using serial sectioning, routine histological staining, and immunohistochemical reactions in human embryos, prefetuses, and fetuses from 8 to 26 gestational weeks. In our study, we used antibodies for tyrosine hydroxylase, dopamine beta-hydroxylase, and phenylethanolamine N-methyltransferase, which are enzymes of catecholamine synthesis, β-III tubulin, and S100. We found two morphological cell types (large and small) in the developing ganglia, organs of Zuckerkandl, and adrenal medulla, and two migration patterns of large cells and small cells. The immunohistochemical characteristics of these cells were determined. We revealed that the number of small cells increased significantly at the ages from 16 to 21–22 gestational weeks, followed by a decrease at 22.5–26 gestational weeks. The presence of two large cell subpopulations was suggested—those that migrate primarily from organs of the Zuckerkandl region and those that differentiate later from the small cells. We also determined that 12 gestational weeks was the age of the first appearance of phenylethanolamine N-methyltransferase reactivity in developing chromaffin cells, temporally correlating with synaptogenesis events. This is important data in the light of the controversial glucocorticoid theory of phenylethanolamine N-methyltransferase induction in humans.

## 1. Introduction

The adrenal medulla (AM), organs of Zuckerkandl (OZ), and other paraganglia (including the carotid body) collectively constitute the sympathoadrenal system. Despite more than a century of research on these organs, their morphogenesis and patterns of interaction during human antenatal development remain poorly understood. This knowledge gap can be attributed to the challenges associated with studying human tissue and the complex evolutionary trajectory of the chromaffin system, which varies significantly across species, thereby limiting the applicability of animal model data to human biology.

The term “OZ” typically refers to the largest collections of chromaffin tissue near the origin of the inferior mesenteric artery. However, the International Federation of Associations of Anatomists recommends the term “para-aortic paraganglia” to describe these structures (also known as the Bodies of Zuckerkandl), effectively eliminating the distinction between the OZ and other paraganglia in humans. Since these organs in humans demonstrate inconstancy in shape, precise location, and poorly defined boundaries between the main bodies and smaller paraganglia, we argue that differentiating the OZ from other paraganglia is an artificial and poorly reproducible categorization. Therefore, in this work, we use the abbreviation “OZ” to refer to all para-aortic paraganglia.

The OZ reaches peak development by the age of 3 years in humans [1], followed by progressive degeneration through mechanisms that are not yet fully understood, potentially involving autophagy [2]. The OZ are thought to be provisional organs, probably involved in the morphogenesis of the AM during the embryonic and fetal periods. Their potential function in human development may include serving as a source of cellular material for the developing AM. In rabbits, for instance, the AM forms through the U-shaped incorporation of the OZ into the cortical anlage, while in rats, the AM is described as developing through a migratory process from the OZ [3].

At the same time, only a few studies on human material have described both extra-adrenal paraganglia and the AM, along with their interactions during embryonic development. One such study is by Hervonen, who identified an invasion route connecting extra-adrenal formaldehyde-induced fluorescing elements with the premedullary area [4]. Molenaar et al. described two distinct cell types in the para-aortic region of a human embryo, referred to as “large” and “small” cells. In a specimen at 9 weeks of gestational age (g.w.), the “large” tyrosine hydroxylase-immunopositive cells were observed traversing the adrenal cortex, whereas the “small” cells were not [5]. This observation raises questions about the ability of partially differentiated cells along the chromaffin pathway to migrate into the cortical anlage. It also remains unclear whether these cells migrate in association with nerves, freely between cortical cells, or through both mechanisms. Kameneva et al. conducted single-cell transcriptomic research on both the human adrenal gland and extra-adrenal paraganglia, revealing different cell lineages and their transitions [6].

Additionally, the role of the OZ is not limited to its morphogenetic function. The OZ may also serve a provisional endocrine role. Their relatively large size compared to the immature fetal AM and their early development of chromaffin characteristics suggest that the OZ could function actively when the AM is not yet fully capable of performing endocrine functions. Specifically, the norepinephrine-synthesizing OZ may serve as effector organs during fetal distress—a role that likely has its roots in their evolutionary history. For instance, there is evidence that norepinephrine is the predominant amine released by chromaffin tissue during hypoxia in hagfishes [7] and, importantly, during fetal distress in humans [8]. This indicates that norepinephrine-storing chromaffin cells in hagfishes play a crucial role in responding to hypoxemia and anoxia. Similarly, it is highly probable that the norepinephrine-containing OZ in humans are key organs in the response to fetal distress.

Another unresolved issue concerning chromaffin tissue is the synthesis of epinephrine. While epinephrine is present in the AM, it is generally absent in extra-adrenal paraganglia, which also lack phenylethanolamine N-methyltransferase (PNMT)—the enzyme responsible for converting norepinephrine into epinephrine. PNMT is a key enzyme in this synthesis process, and numerous studies have shown that glucocorticoids stimulate PNMT expression in chromaffin cells [9,10,11,12]. This has led to the hypothesis that a portal system within the adrenal gland might exist, allowing higher concentrations of glucocorticoids to reach the AM [13]. However, this idea remains controversial, as no such portal system has been observed in rats [14]. In contrast, in hagfishes, chromaffin cells have been shown to be insensitive to glucocorticoids [15] and have minimal contact with interrenal cells. Despite this, they are still capable of synthesizing PNMT and epinephrine. This raises questions about the evolutionary aspects of the morpho-functional integration of chromaffin tissue with the adrenal cortex, which remain unclear. Thus, considering the evolutionary development of the sympathoadrenal system and its reflection in human ontogenesis, we believe that studying the comparative morphological characteristics of the key organs in this system during antenatal development is most productive.

Although several studies have investigated molecular genetic aspects of human OZ and AM development [6,16], they lack comprehensive data on spatio-temporal distribution patterns of differentiating chromaffin cells. This gap hinders interpretation within the broader context of the para-aortic region and the mutual interactions between these structures. A critical outstanding question involves the organizational architecture of these cells, particularly their microstructural arrangements and emergent tissue patterns. For these reasons, classical histological methods combined with immunohistochemistry retain fundamental importance, even amidst advances in single-cell transcriptomics and other molecular methods. This is particularly relevant given that the presence of transcripts in cells does not necessarily correlate with the presence of functional protein product [17]. Moreover, the presence of a particular protein in a cell does not define the cell type because cell type specification requires combined analysis of molecular signatures and morphological characteristics.

In this study, we address these knowledge gaps by systematically characterizing cytological, histological, and immunohistochemical features of the developing human sympathoadrenal system, with particular focus on spatiotemporal dynamics of chromaffin cell distribution in OZ and AM.

## 2. Materials and Methods

### 2.1. Antenatal Material

The investigation was carried out on human embryos, prefetuses, and fetuses (for details, see Appendix A). During sample collection, we recorded the gender, gestational age, clinical diagnosis of the mother and fetus, and the causes of pregnancy termination or fetal death. When clinical data were insufficient or unavailable, fetal age was determined according to the criteria established by Milovanov and Saveliev [18]. The study was conducted on autopsy samples from the Collection of the Laboratory of Nervous System Development at the Avtsyn Research Institute of Human Morphology, Petrovsky National Research Centre of Surgery, Moscow, Russian Federation. Tissue specimen collection and handling adhered to Russian legislation and the Declaration of Helsinki. Consent for the use of human tissue in research was obtained from the legal representatives of all subjects involved in the study. All protocols were approved by the local Ethics Committee of the Research Institute of Human Morphology (protocol No. 33 (9), 7 February 2022).

In fetuses and late prefetuses, the adrenal glands and aorta, along with surrounding tissues including the OZ, were taken separately. For embryos, the entire body was fixed and embedded, followed by serial sectioning. In early prefetuses, the adrenal glands were either taken as part of an organ complex with the aorta and surrounding tissues or, depending on the size of the organ complex, one adrenal gland was taken separately.

### 2.2. Fixation and Processing

All the material was fixed in 10% buffered formalin (BioVitrum, St. Petersburg, Russia), dehydrated using a standard tissue protocol with IsoPrep (BioVitrum, St. Petersburg, Russia), and embedded in Histomix (BioVitrum, St. Petersburg, Russia). Serial sections (thickness = 5 μm) were taken using a Leica RM2245 microtome (Leica Biosystems, Nussloch, Germany). Every 20th section was deparaffinized and routinely stained with hematoxylin and eosin. The sections were examined under Leica DM2500 and Leica DM2000 light microscopes (Leica Microsystems, Wetzlar, Germany).

### 2.3. Immunohistochemistry

The most representative sections were selected for immunohistochemical (IHC) analysis. These sections were deparaffinized, rehydrated, and subjected to antigen retrieval by boiling in citrate buffer (pH 6.0) for 5 min. For immunohistochemistry using antibodies against tyrosine hydroxylase and β-III tubulin, the antigen retrieval step was omitted. However, for PNMT antibodies, the sections were boiled in citrate buffer for an extended period (30 min). Next, the sections were treated with a 3% H2O2 peroxide solution for 10 min to block endogenous peroxidase activity. They were then treated with Ultra V Block (Thermo Fisher Scientific, Waltham, MA, USA) for 5 min. After blocking, the specimens were incubated with primary antibodies for 60 min at room temperature (see Table 1 for details). We also utilized 200 kDa neurofilaments (murine monoclonal, Merck, Darmstadt, Germany). However, we were unable to obtain stable results with the antibodies for neurofilaments, likely due to the instability of the target antigen [19]. The UltraVision Quanto Detection System kit (Thermo Fisher Scientific, Waltham, MA, USA) was used as the detection system.

The sections were photographed with Leica Flexacam C3 (Leica Microsystems, Wetzlar, Germany). The digital images were saved in TIFF format and matched for brightness and contrast using Adobe Photoshop CC 2019 (Adobe Systems, Inc., San Jose, CA, USA).

### 2.4. Morphometry and Statistics

In the developing AM two cell types were detected morphologically: large cells and small cells (for details, see Results). The most representative AM-containing sections in terms of volume were stained with antibodies for TH and photographed in their entirety using Leica Flexacam C3 (Leica Microsystems, Wetzlar, Germany) with a 10× objective. The digital images were saved in TIFF format.

Morphometry was performed using the program ImageJ 1.54p. The number of large cells and small cells was counted. The ratio of large cells to small cells was also calculated.

Three distinct age groups were identified by morphological and morphometrical patterns. The first age group contained 6 embryos and prefetuses from 8 to 12 g.w. (No. 1–6). The second age group consisted of 6 fetuses from 16 to 21–22 g.w. (No. 7–12). The third age group contained 2 fetuses from 22–23 g.w. to 25–26 g.w. (No. 13–14).

Since there were only two cases in the third age group, statistical analysis was performed only for the first and the second age groups. The statistical significance of the results was assessed using Statistica 10.0 software (StatSoft, Tulsa, OK, USA). Three parameters of these two groups were analyzed: number of large cells, number of small cells, and the ratio of large cells to small cells.

The significance of differences was assessed using the nonparametric Mann–Whitney test. A stronger Bonferroni-corrected *p*-value test was used. It was obtained by following formula:$$p_{*}=p÷3$$

Given that *p* = 0.05, the differences were considered significant at $p_{*}<0.016$.

## 3. Results

### 3.1. General Description

At all investigated stages in the abdominal region, the cortical anlagen of the adrenal glands were already present as oval-shaped conglomerates of large cells with abundant oxyphilic cytoplasm (Figure 1A,C). Nearby, in the para-aortic region adjacent to the aorta, large conglomerates of cells representing the developing ganglia and OZ anlage were observed, along with crossing nerve bundles. In the early stages, the boundaries between the ganglionic anlagen and the OZ anlage were indistinct (Figure 1A). However, by approximately 12 g.w., a prominent thin capsule surrounding the OZ anlage became apparent (Figure 1C). From that moment onward, we can refer to this structure as the mature OZ.

At early developmental stages (from 8 g.w. to 12 g.w.), the developing ganglia and OZ anlage consisted of two morphological cell types. The first and the most common cell type was the small cell type (SC). These cells had scarce, sometimes inconspicuous cytoplasm and hyperchromatic nuclei, appearing as small blue cells that formed rounded or oval solid structures (Figure 2A,B and Figure 3A,B). At later stages (from about 12 g.w.), numerous neuron-like cells with processes and conspicuous large nuclei with nucleoli began to appear among the SC conglomerates, indicating differentiation along the neuronal pathway (Figure 4A–C).

The second cell type was the large cell type (LC). LCs had clear oxyphilic cytoplasm and clear prominent nuclei with nucleoli. At early stages, they formed small rounded clusters (Figure 2A,B and Figure 3A,B), which increased rapidly with age. By 12 g.w., these LCs had already formed well-developed, properly shaped, and functionally active OZ (Figure 1C,D and Figure 4A–D). Such rapid growth in size of these structures could be explained either by proliferative activity of LCs themselves, or differentiation of a part of SCs along the chromaffin lineage.

LCs and SCs correspond to the “large” and “small” cells described by Molenaar et al. [5]. The morphological description of LCs also aligns with the characteristics of AM chromaffin cells, which have large nuclei with “open chromatin,” as reported by Cooper et al. [20].

SCs persisted in the adrenal anlage even at the latest investigated age (25–26 g.w.) (Figure 5B), contrasting with their absence in both developing ganglia and OZ regions at this developmental period.

During the investigated stages of human ontogenesis, the structures formed by these cells underwent distinct morphological and immunohistochemical changes as follows. Results of immunohistochemical reactions are shown in Appendix A.

### 3.2. 8–9 Gestation Weeks

At 8–9 g.w., para-aortic conglomerates consisted of rounded-to-oval clusters of SCs and LCs. Numerous mitotic figures were present among SCs, with occasional mitoses observed in LC clusters. The majority of SCs were found in contact with nerve bundles running along the aorta or extending medially toward the adrenal anlage. LC clusters were located close to SC conglomerates and were also connected to nerve bundles. Immunohistochemically, LCs displayed strong cytoplasmic reactivity for TH, DBH, and βIII-tubulin (Figure 1A,B, Appendix A). In contrast, SCs showed significantly weaker reactivity with antibodies for TH and DBH, with some SCs being negative for DBH. However, SCs exhibited marked cytoplasmic expression of βIII-tubulin. Neither LCs nor SCs expressed PNMT at this developmental stage, suggesting that extra-adrenal LCs may already be producing norepinephrine, but not epinephrine. S100 immunoreactivity was weakly positive in the nuclei and cytoplasm of occasional cells associated with nerves. In specimen No. 3, S100+ cells were also observed at the periphery of both LC and SC clusters.

In the cortical anlagen of the adrenal glands, we identified individual cells and small groups of cells that were morphologically and immunohistochemically similar to LCs observed in extra-adrenal conglomerates. These cells were dispersed among the oxyphilic cells of the cortical anlage, demonstrating a distinct distribution gradient-their density progressively decreased with increasing distance from extra-adrenal LC aggregates. This spatial pattern strongly suggests a migratory process, indicating that large TH+DBH+βIII+ cells may migrate from the extra-adrenal chromaffin tissue to the cortical anlagen. This observation aligns with the findings of Hervonen [4] and Molenaar et al. [5]. βIII-tubulin positive nerve fibers were also seen transitioning adrenal anlage, with some of these fibers closely associated with LCs interspersed within the cortical anlage.

### 3.3. 11–12 Gestation Weeks

At 11–12 g.w., both large (LCs) and small cells (SCs) persisted, with increasingly distinct migratory patterns of LCs from the OZ anlagen toward the cortical anlage becoming evident (Figure 1D). In some areas, these migrating LCs appeared to travel along βIII+ nerves, while in others, they were freely interspersed among cortical cells (Figure 2A,B and Figure 3A,B). Intra-adrenally, LCs were strongly positive for TH, DBH, with slightly weaker positivity for βIII-tubulin. Clusters of SCs were also present intra-adrenally, but they were less abundant, and their migratory pathways appeared less distinct compared to those of LCs. SCs were strongly positive for βIII-tubulin and weakly positive for TH and DBH.

Extra-adrenally, at the level of the developing kidneys, large rounded-to-oval masses of the developing OZs, composed primarily of LCs, were surrounded by clusters of SCs, which were mostly located peripherally. The boundaries between these SC and LC clusters appeared more distinct in one 12 g.w. specimen (No. 6) (Figure 1C) compared to earlier specimens (Nos. 4 and 5) (Figure 2A,B), with a thin capsule already forming around the developing OZ. However, the peripheral location of SCs was not consistent, as in other sections and specimens, SCs and LCs formed independent round structures composed of only one cell type.

Extra-adrenally, at the level of the developing kidneys, large rounded-to-oval masses of the developing OZ, composed primarily of LCs, were surrounded by peripheral clusters of SCs. The boundaries between SC and LC clusters appeared more distinct in a 12 g.w. specimen (No. 6; Figure 1C) compared to earlier specimens (Nos. 4 and 5; Figure 2A,B), with an incipient capsule forming around the OZ. However, this organizational pattern showed variability, as some sections revealed independent round structures containing exclusively either SCs or LCs.

As in previous stages, extra-adrenal LCs and, to a lesser extent, SCs were positive for TH and DBH. Given the large size of the developing OZs formed by LCs, these organs were likely the main source of norepinephrine in the developing human organism.

Among the extra-adrenal SCs, a few scattered, larger, triangle-shaped cells with more prominent cytoplasm and large nuclei with conspicuous nucleoli were observed, likely representing developing neurons.

S100-positive cells with marked nuclear and cytoplasmic expression were found along the nerves, at the periphery of LC conglomerates, and among SCs (Figure 5C). These S100-positive cells were round, oval, or spindle-shaped, with no morphological differences based on their location along the nerve or at the periphery of LC structures.

In one of two specimens at 12 g.w. (No. 6), intracortical chromaffin cells exhibited strong cytoplasmic PNMT immunoreactivity (Figure 5A), while in extra-adrenal chromaffin tissue and in the other specimen of similar age (No. 5), PNMT was negative.

### 3.4. 16 Gestation Weeks

At 16 g.w. (1 specimen), no significant differences were observed compared to the 12 g.w. prefetus. However, LCs appeared negative for PNMT. This may be due to the high sensitivity of PNMT to autolytic changes in tissues, which leads to result instability. Other cytological and immunohistochemical characteristics of large and small intra- and extra-adrenal cells remained consistent with those observed at earlier stages.

### 3.5. 20–22 Gestation Weeks

At 20–22 g.w. intra-adrenally, numerous large, rounded, and oval groups of SCs were observed within the cortical tissue and occasionally in the walls of vessel-like structures, directly adjacent to the lumens. Some lumens contained numerous erythrocytes (Figure 2C). This pattern of SC location within true vessel lumens or vessel-like structures may represent either artificial changes during specimen preparation or a true anatomical feature. Regardless, it was a consistent morphological characteristic in all specimens at this stage. Such vessel-like spaces correspond to previously described cell-free cavities [21,22]. Iwanaga and Fujita suggested that these structures might result from degeneration of so-called primitive sympathetic cells, which we refer to as SCs here. However, we observed no evidence of necrosis or apoptosis [22].

Intra-adrenal LCs were organized in small nests, primarily located at the periphery of SC groups and among cortical cells. LCs exhibited strong TH+DBH+ and weak βIII+ or βIII− immunoreactivity (Figure 2D and Figure 3C,D). This weaker positivity of LCs for βIII-tubulin aligns with the findings of Katsetos et al. [23], who demonstrated focal distribution of βIII positivity in chromaffin cells in human fetuses at 20 g.w. and later. LCs were also strongly positive for PNMT, except for three of seven specimens, which appeared to be negative due to the high sensitivity of this IHC marker. SCs were strongly βIII+, and weaker TH+DBH+ reactivity. S100+ cells were present in the vessel walls, at the periphery of SC clusters, and as individual cells among LC groups (Figure 5D). These S100+ cells were spindled or oval in shape.

Extra-adrenally, well-formed, rounded or oval-shaped ganglia composed of large triangle-shaped stellate neurons with prominent processes and nucleoli were observed (Figure 4A,B). Extra-adrenal SCs were few. Neurons in the ganglia were βIII+ and, to a lesser extent, TH+DBH+. In close proximity to ganglia, large OZs represented by LCs were located. They were strongly TH+DBH+ and, to a lesser extent, βIII+ (Figure 4A,B). Nerve fibers, connecting ganglia and paraganglia, were also βIII+. S100+ cells were present along nerves and among large cells in OZ (Figure 4D).

### 3.6. 22.5–26 Gestation Weeks

At this stage, intra-adrenally, both LCs and SCs remained present, with occasional neuron-appearing cells with processes and prominent nucleoli observed among SCs. Extra-adrenally, SCs were not detected. Other morphological and immunohistochemical features remained similar to those at earlier stages (Figure 2F, Figure 3E,F, Figure 4C and Figure 5B).

### 3.7. Morphometry and Statisctics

Preliminary microscopic investigation revealed visible changes in the numbers of LCs and SCs during embryonic morphogenesis. Taking into account the parameters of changes in these cells, three age periods were identified: from 8 to 12 g.w. (group 1), from 16 to 21–22 g.w. (group 2), and from 22–23 g.w. to 25–26 g.w. (group 3).

To quantify the number of cells and their ratio more accurately, a morphometric study was conducted. The results are presented in Table 2.

It was found that the number of LCs had a slight tendency to increase with gestational age. However, differences between group 1 and 2 were not statistically significant (*p* = 0.378) (Figure 6A). At the same time, the number of SCs was significantly higher in group 2 than in group 1 (*p* = 0.005) (Figure 6B). The L/S ratio demonstrated the opposite trend, which was also statistically significant (*p* = 0.005) (Figure 6C).

Since there were only two cases in the third age group, statistical analysis was performed only for the first and second age groups. However, the number of SCs visually decreased, while the LCs and L/S ratio in the third age group increased (Figure 2E).

## 4. Discussion

Chromaffin tissue has undergone significant evolutionary changes from cyclostomes to placental mammals. In cyclostomes, this tissue is represented by scattered stellate cells within the walls of the cardinal vein, aorta, and its branches, and in the endocardium of the heart. Cyclostomes possess a poorly developed autonomic nervous system, and in this group, chromaffin tissue generally operates independently of nervous control [3]. For example, in hagfishes, chromaffin tissue is known to lack preganglionic innervation [24,25]. Over the course of evolution, chromaffin cells have lost their independence from the nervous system and acquired innervation. Simultaneously, these cells consolidated into a distinct organ. This development improved the regulatory mechanisms of chromaffin cell endocrine function. As a result, placental mammals now possess a well-developed, richly innervated AM as part of the adrenal gland, a complex endocrine organ. The AM became an integral component of the mammalian sympathoadrenal system, while most paraganglia have become nearly non-functional rudiments. However, some paraganglia are believed to have acquired specific functions during phylogenesis. For instance, the carotid body developed chemosensitive function [26]. As demonstrated in our study, other paraganglia, such as the OZ, play an important role in human embryological and fetal development. Thus, the OZ function not only as endocrine provisional organs but also as a morphogenic factor for the AM.

Our results suggest two potential pathways for AM cell population recruitment during embryogenesis. The first and most prevalent pathway involves LC migration from OZ anlagen. The second, less histologically apparent pathway is associated with the migration of SCs from extra-adrenal conglomerates.

At the earliest investigated stage (8–9 gestational weeks), chromaffin cells (indicated here by LC morphology and marked cytoplasmic TH+ and DBH+ immunoreactivity) were already present in the extra-adrenal para-aortic region. These developing chromaffin cells were also observed in the medial part of the cortical anlage, either freely interspersed among cortical cells or in connection with nerves, resembling a migratory pattern. This suggests that the OZ anlagen may play a crucial role in the distribution of partially differentiated (TH+, DBH+) chromaffin lineage cells, contributing to the formation of the AM. Alternatively, it is possible that even at the earliest investigated stage (8–9 g.w.), we captured not the ongoing migration, but the result of previous migration transitionally through the para-aortic region with subsequent differentiation of cells along the chromaffin pathway.

At the earliest investigated stages, SCs were primarily located extra-adrenally in close association with nerve bundles, with only occasional SCs observed among cortical cells. The weak SCs positivity for antibodies could have contributed to difficulties in SCs detection. As a result, their migration and presence in cortical anlage were less apparent compared to the more evident migratory pattern of LCs.

At later stages, the intra-adrenal LC migratory pattern became more obvious. Alongside LCs, small groups of SCs were also observed intra-adrenally. These SCs were typically found in close contact with nerves, whereas LCs were often seen within the cortical anlage, independent of nerve structures. This suggests that LCs may migrate not only along nerves but also freely between cortical cells, while SCs primarily migrate to the adrenal anlage along nerve pathways.

At the same time, some LCs were located at the periphery of SC clusters, closely associated with them. This suggests that a portion of SCs may differentiate into LCs. Moreover, at the age of 22.5–26 g.w., the ratio of LCs to SCs visually increased, likely indicating that the majority of SCs had differentiated into LCs and, to a lesser extent, into neurons. However, more samples are needed to confirm this.

Placing our findings in the context of recent single-cell transcriptomics work [6], we identified some correlations with their data. According to Kameneva’s work, there are two ways of chromaffin cell differentiation. The first one is a direct transition from Schwann cell precursors (SCPs) to chromaffin cells. These chromaffin cells correspond to the primarily migrated LCs we observed. The second one is a two-step transition from SCPs via intra-adrenal sympathoblasts (IASs) to chromaffin cells. These IASs correspond precisely to the SC populations identified in our study. Kameneva et al. classified the clusters of SCs with LCs at the periphery as ganglia-like structures (GLS) following the idea that these structures predominantly give rise to intra-adrenal ganglia. They justify this by the discovery of intra-adrenal ganglia in the postnatal period.

However, our analysis of later fetal stages demonstrates that only a minority of these so-called ganglia-like structures differentiate into ganglia. First of all, it is proved by non-ganglion morphology obtained by these clusters at later stages, which is characterized by poorly defined borders and vessel-like spaces. Secondly, we observed the discrepancy between abundant SCs in our second age group and the scarcity of ganglia in the third age group and mature adrenal glands, which was demonstrated by other researchers [20]. In addition, the postnatal ganglia described by Kameneva et al. were positive only for chromogranin A and vasoactive intestinal peptide, but not for tyrosine hydroxylase, although cells associated with GLS in the antenatal period were strongly positive for it.

This raises questions about the role of the initially migrating LCs, which are not connected to the differentiation of intra-adrenal SC clusters. One possibility is that these primarily migrating LCs play a provisional role in producing catecholamines and facilitating the migration of SCs into the cortical anlage. Following this, the primarily migrated LCs may undergo apoptosis. Another possibility is that both the initially migrating LCs and those differentiated from SCs collectively contribute to the formation of the AM (Figure 7). Additionally, the specific roles of these two cell types in AM morphogenesis remain uncertain. At the early stages, the developing AM was primarily composed of migrating LCs, with only occasional SCs present in the cortical anlage. However, at later stages, large clusters of SCs were observed within the adrenal gland. This may be attributed to the high proliferative activity of SCs, which likely contributes to the significant increase in the overall cell population.

The last interesting observation is related to the peculiarities of PNMT synthesis. At the age of 12 g.w., we observed strong cytoplasmic positivity for PNMT in intra-adrenal LCs, both those associated with SCs and those lying freely. This finding is intriguing in light of ultrastructural results reported by Hervonen [4], who identified the first ultrastructural features of epinephrine-containing granules at 16 g.w. in the human fetus AM, approximately four weeks after the appearance of PNMT in our study. Hervonen also noted that typical synaptic profiles were present on medullary chromaffin cells in a 12-week-old fetus but stated that mature synapses on these cells did not appear until 14 weeks. In contrast to intra-adrenal chromaffin cells, Hervonen demonstrated that extra-adrenal paraganglia remained non-innervated, as no synapses were found on their chromaffin cells, and they lacked epinephrine-containing granules.

These observations, along with our results, suggest a correlation between the presence of synapses on chromaffin cells and the presence of epinephrine-containing granules, and consequently, the presence of PNMT, the enzyme responsible for epinephrine synthesis. Therefore, it is possible that PNMT synthesis in humans is connected to the process of synaptogenesis. Moreover, it is probable that the establishment of synapses triggers the further differentiation of chromaffin cells from norepinephrine-synthesizing to epinephrine-synthesizing cells. This idea, that synaptogenesis can induce differentiation, dates back to an early study by Golub [21]. This hypothesis is also in accordance with widely known facts about the influence of synaptic activity on the expression of some genes. The most well-known example is neurotrophic effects on muscles [27] and some effects on neurons in the brain [28,29]. Once a synaptic terminal appears on a cell, it begins to function, since there is no structure without function [21]. This functioning synapse can cause a shift in the cellular program, after which the expression of PNMT begins.

However, validating this hypothesis through existing literature presents significant challenges. For example, the temporal relationship between synapse formation and PNMT expression remains unclear in rats. Ultrastructural features of mature synapses on chromaffin cells in rats appear at 15.5 days of gestation, suggesting that neural control over medullary chromaffin cells could be established from this point [30]. However, functional studies show that in 1-day-old newborn rats, neural regulation of catecholamine release is absent until about 8 days of postnatal life [31], even though epinephrine is already present in medullary chromaffin cells. Critically, this developmental dissociation between structural synaptogenesis (day 15.5) and functional neural control (day 8) does not preclude synaptic influence on PNMT induction. Nevertheless, these apparent discrepancies highlight the need for further investigation.

The appearance of PNMT at 12 gestational weeks (g.w.) in a human prefetus raises questions about the role of glucocorticoids in the expression of PNMT. The adrenal cortex in human fetuses is distinct from that in other species, and cortisol is believed to be produced transiently early in gestation (around 7–10 g.w.). However, due to the lack of hydroxy-delta-5-steroid dehydrogenase, 3 beta- and steroid delta-isomerase 2 activity—the enzyme essential for de novo cortisol synthesis—cortisol biosynthesis appears to be suppressed until 23–24 g.w. [32]. Consequently, at 12 g.w., the hypothetical level of cortisol would be low, making it unlikely to stimulate PNMT expression. This suggests that glucocorticoid induction of PNMT synthesis in humans may not be the primary mechanism. However, additional human samples are needed to prove that our result is not connected with some individual variability, and further investigation into the adrenal cortex, adrenal medulla, and extra-adrenal paraganglia as a complex is necessary to understand this issue in humans.

## 5. Conclusions

In the current study, we investigated the developmental parallels between the AM and OZ in human embryos and fetuses at different stages of development. We found that the OZ may not only function as a provisional norepinephrine-synthesizing organ, but also OZ anlage plays a crucial role in the morphogenesis of the human adrenal gland, serving as a distributor of chromaffin cells that migrate to the cortical anlage. Two morphological cell types, LCs and SCs, were identified in the developing AM, extra-adrenal ganglia, and OZ, and the dynamics of changes in their amounts and ratios were established. The presence of two LC subpopulations was also suggested—those that migrate primarily from OZ region and those that differentiate later from the SCs.

Besides, we observed that PNMT first appeared at 12 g.w. of human development–a finding that warrants further investigation in relation to the processes of synaptogenesis in the AM. A deeper analysis and comprehensive study of the developing chromaffin and cortical tissues could provide insights into the evolutionary significance of the coalescence of these two distinct tissues.

## Figures and Tables

**Figure 1 life-15-01214-f001:**
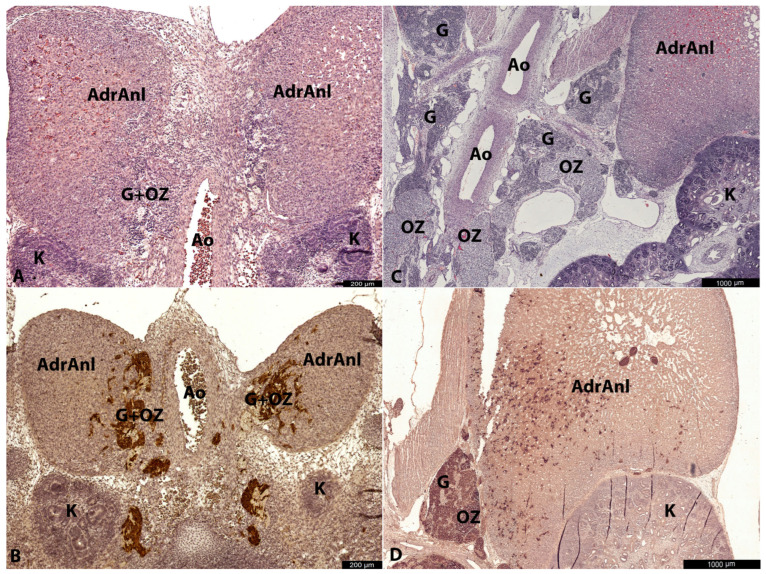
Overview pictures of the paraadrenal region in the embryo and prefetus. Ao—aorta, AdrAnl—adrenal anlage, G—ganglion, OZ—organs of Zuckerkandl, K—kidney. The upper line shows sections stained with hematoxylin and eosin, and the lower line is IHC with TH. At 8–9 g.w. (No. 1) human embryo, overview. Ganglia and OZ are located in close proximity to adrenal anlages and aorta (**A**). At 8–9 g.w. (No. 1) human embryo, overview. TH+ cells of developing ganglia and OZ infiltrate adrenal anlages (**B**). At 12 g.w. human prefetus (No. 6), para-adrenal region, overview picture. Adrenal medulla is barely seen in the cortical anlage, while OZ are prominent, well-defined structures (**C**). At 12 g.w. human prefetus (No. 6), paraadrenal region, immunohistochemistry with TH. Adrenal medulla is represented by scattered TH+ cells and their groups, and a prominent gradient of TH+ cells is seen at a distance from OZ, with predominance of TH+ cells in the medial part of the adrenal anlage (**D**).

**Figure 2 life-15-01214-f002:**
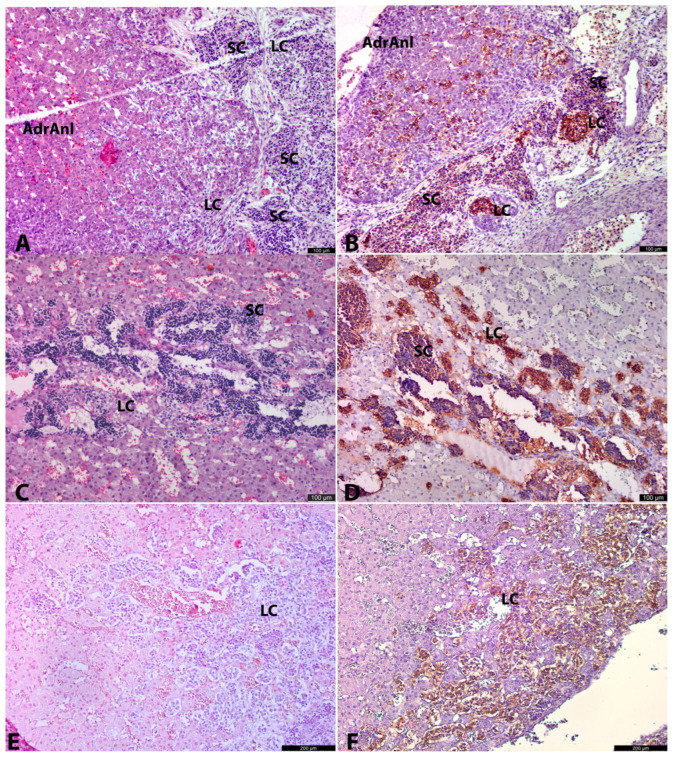
Adrenal anlages in 3 age groups. AdrAnl—adrenal anlage, LC—large cells, SC—small cells. The left column of micrographs shows sections stained with hematoxylin and eosin. The right column shows IHC with TH, 11–12 g.w. human prefetus (No. 4), adrenal anlage with adjacent tissues. Large clear cells (LCs) are seen intermingling between oxyphilic cortical cells. Conglomerates of SCs and LSc (to a lesser extent) are located in close proximity to the adrenal anlage. Nerve fibers are connecting small cell structures and invading cortical anlage (**A**). At 11–12 g.w. human prefetus (No. 4), adrenal anlage with adjacent tissues. LCs both inside and outside the adrenal anlage are highly positive for TH. SCs are weakly positive for TH (**B**). At 20–21 g.w. human fetus (No. 10), adrenal anlage. Abundant rounded clusters of SCs are seen along with peripherally located LCs. A lot of vessels and/or vessel-like spaces/cavities or “cisterns” are found between small and large cell groups. In many of the lumens of these vessels/vessel-like spaces erythrocytes are seen (**C**). At 20–21 g.w. human fetus (No. 10), adrenal anlage. Rounded clusters of SCs with weak cytoplasmic TH positivity are seen along with intensively positive groups of LCs. “Cisterns” in the SC groups are also seen (**D**). At 25–26 g.w. human fetus (No. 14), adrenal gland. LCs are the predominant cell type of the AM (**E**). At 25–26 g.w. human fetus (No. 14), adrenal gland. LCs are highly positive for TH (**F**).

**Figure 3 life-15-01214-f003:**
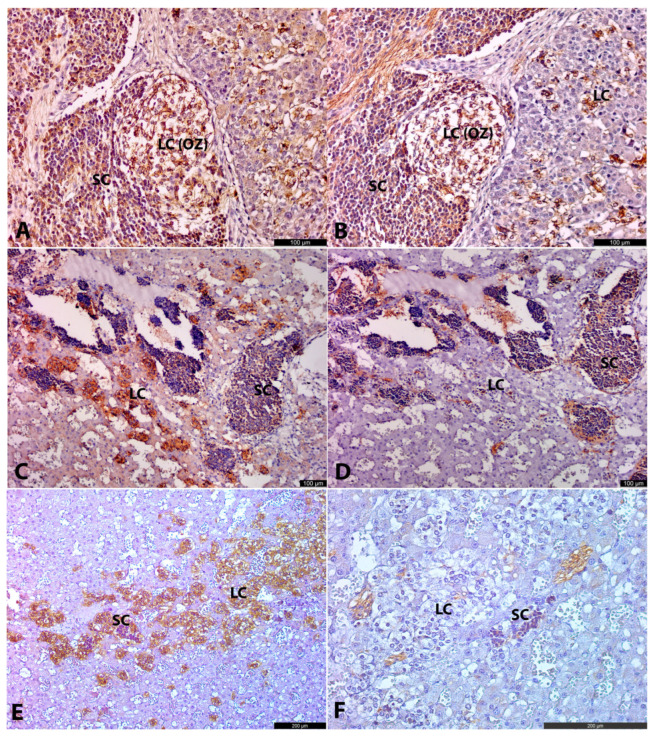
Adrenal anlages in 3 age groups. LC—large cells, SC—small cells, OZ—organs of Zuckerkandl. Left column shows IHC with DBH, right column demonstrates IHC with βIII-tubulin. At 11–12 g.w., human prefetus (No. 5). At the right part of the picture adrenal anlage is seen with DBH+ LCs intermingling between cortical cells. At the left part OZ anlage consisting of DBH+ LCs is seen surrounded by SCs of the ganglion anlage with weaker DBH positivity (**A**). At 11–12 g.w., human prefetus (No. 5). At the right part of the picture, the adrenal anlage is seen with βIII+ LCs intermingling between cortical cells. At the left part, OZ anlage consisting of βIII+ LCs is seen surrounded by βIII+ SCs of the ganglion anlage. Nerve fibers between clusters of SCs are highly positive for βIII+ (**B**). At 20–21 g.w. human fetus (No. 10), adrenal anlage. Rounded clusters of SCs with weak cytoplasmic DBH positivity are seen along with intensively positive groups of LCs. A lot of vessels and/or vessel-like spaces/cavities or “cisterns” are found between small and large cell groups. In many of the lumens of these vessels/vessel-like spaces, erythrocytes are seen (**C**). At 20–21 g.w. human fetus (No. 10), adrenal anlage. Rounded structures of SCs with strong cytoplasmic βIII positivity are seen. βIII+ are found connecting small and large cell groups. LCs have weak or no reactivity with βIII (**D**). At 22–23 g.w., human fetus (No. 13), adrenal gland. Highly positive for DBH LCs are the predominant cell type. Few clusters of SCs with weaker DNH positivity are seen among LCs (**E**). At 22–23 g.w., human fetus (No. 13), adrenal gland. βIII+ nerve fibers among clusters of LCs and SCs are seen. LCs are negative for βIII+, while SCs demonstrate βIII+ cytoplasmic positivity (**F**).

**Figure 4 life-15-01214-f004:**
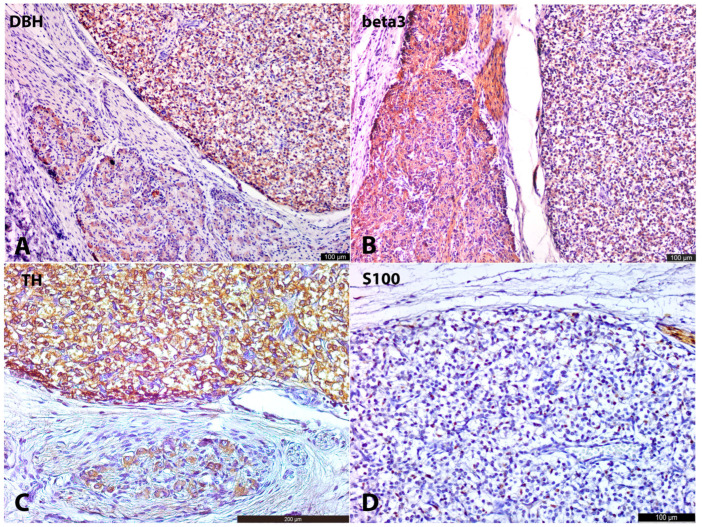
Organs of Zuckerkandl, IHC. At 20–21 g.w. human fetus (No. 10), IHC with DBH. The left corner is occupied by the ganglion with weakly DBH+ neurons. The right upper corner is occupied by OZ with DBH+ LCs (**A**). At 20–21 g.w. human fetus (No. 10), IHC with βIII. Neurons in the ganglia (left part of the picture) are highly βIII+ along with nerve fibers. OZ (right part of the picture) are weakly positive for βIII (**B**). At 22–23 g.w. human fetus (No 13), IHC with TH. Neurons in the ganglia (lower part of the picture) are positive for TH. LCs in the OZ (upper part) are strongly positive for TH (**C**). At 20–21 g.w. human fetus (No 10), OZ, IHC with S100. S100+ cells (nuclear and cytoplasmic staining) are scattered among the LCs of OZ (**D**).

**Figure 5 life-15-01214-f005:**
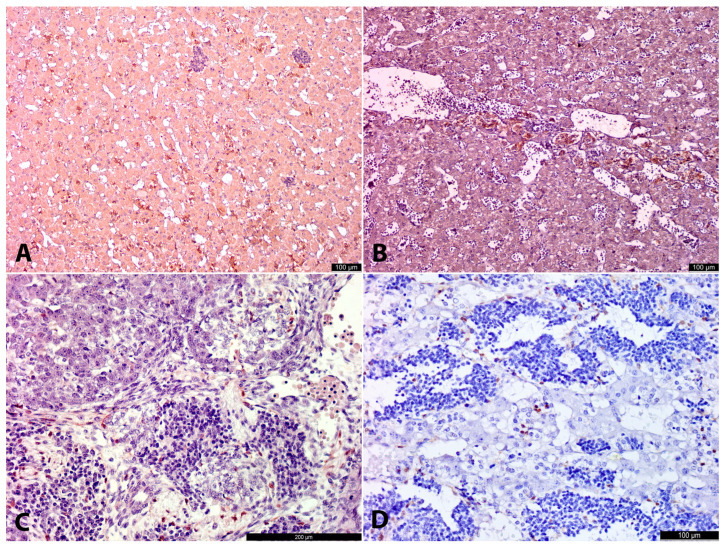
Adrenal anlages, PNMT and S100 IHC. At 12 g.w. human prefetus (No. 6), adrenal anlage, IHC with PNMT. PNMT+ cells are located among cortical cells. SCs are seen as PNMT−“blue” cell clusters surrounded by cortical cells (**A**). At 25–26 g.w. human fetus (No. 14), IHC with PNMT. LCs are strongly PNMT+, while SCs are negative (**B**). At 11–12 g.w. (No. 4), IHC with S100. The upper corner shows adrenal anlage with a cluster of LCs with scattered S100+ cells. The lower corner demonstrates LC round conglomeration with a few S100+ cells. The LC cluster is surrounded by SCs of the ganglion anlage and connected with the nerve with the LC cluster in the adrenal anlage (**C**). At 20–21 g.w. human fetus (No. 10), IHC with S100. S100+ cells surround clusters of SCs and LCs (**D**).

**Figure 6 life-15-01214-f006:**
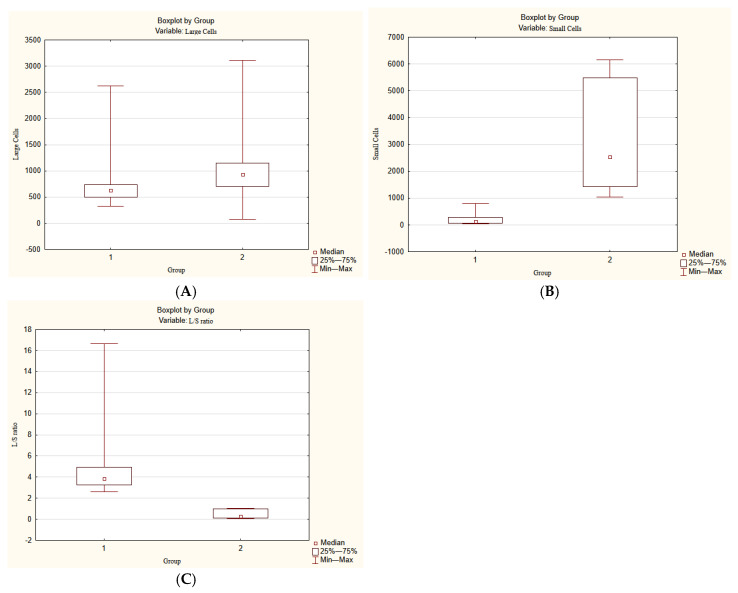
Statistical differences in the number of LCs: (**A**) (*p* = 0.378), the number of SCs, (**B**) (*p* = 0.005), and the LC/SC ratio, (**C**) (*p* = 0.005) between group 1 and group 2.

**Figure 7 life-15-01214-f007:**
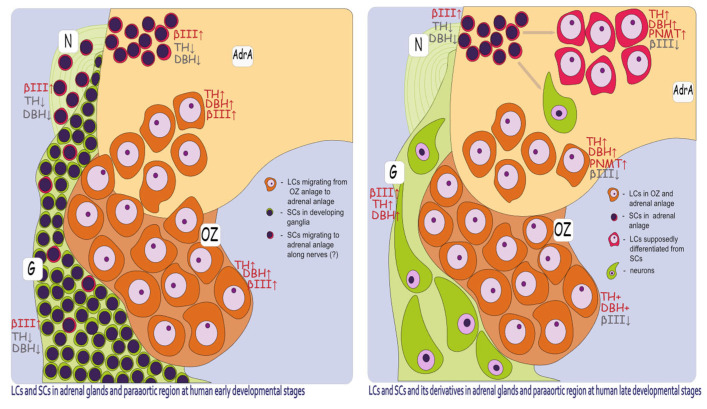
Schematic representation of migration and differentiation of LCs and SCs and their immunohistochemical characteristics. AdrA—adrenal anlage, G—ganglion, OZ—organs of Zuckerkandl, N—nerve. ↑—strong reaction with antibodies, ↓—weak reaction with antibodies. At the early human developmental stages, LCs (TH↑DBH↑βIII↑) migrate from the OZ anlages to the adrenal anlage. SCs (βIII↑TH↓DBH↓) are mostly located in the developing ganglia with some of them travelling along nerves into the adrenal anlage. At the later developmental stages, SCs proliferate and differentiate into LCs, which, along with primarily migrated LCs, make up the two subpopulations of the AM. Some of the SCs differentiate into neurons of the AM.

**Table 1 life-15-01214-t001:** Primary antibody characteristics.

No.	Antigen, Host Species, Supplier	Working Dilution
1	Tyrosine hydroxylase (TH), rabbit polyclonal. Abcam (Cambridge, UK)	1:200
2	Dopamine beta-hydroxylase (DBH), rabbit polyclonal CSB-PA020742. Cusabio (Houston, TX, USA)	1:400
3	Phenylethanolamine N-methyltransferase (PNMT), rabbit anti-Homo sapiens (Human) polyclonal CSB-PA02939A0Rb. Cusabio (Houston, TX, USA)	1:300
4	β-III tubulin, rabbit polyclonal. Abcam (Cambridge, UK) (βIII)	1:500
5	S100, rabbit polyclonal. Thermo Fisher Scientific (Waltham, MA, USA)	1:1000

**Table 2 life-15-01214-t002:** Number of LC, SC, and L/S ratio in three age periods.

Case	Large Cells	Small Cells	L/S Ratio	GW	Group
1	321	65	4.93846154	8–9	1
2	710	273	2.6007326	8–9	1
3	552	129	4.27906977	9	1
4	733	44	16.6590909	11–12	1
5	501	147	3.40816327	11–12	1
6	2618	798	3.28070175	12	1
7	74	1422	0.0520393812	16	2
8	702	5484	0.128008753	20	2
9	1088	1050	1.03619048	20	2
10	1151	6149	0.187184908	20–21	2
11	785	1968	0.398882114	21	2
12	3112	3121	0.997116309	21–22	2
13	1753	718	2.44150418	22–23	3
14	5302	1125	4.71288889	25–26	3

## Data Availability

All data presented in this study are available in this article and Appendix A.

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
