# Peer review of "Developmental Parallels Between the Human Organs of Zuckerkandl and Adrenal Medulla"

_life, 2025, doi:10.3390/life15081214_

Round 1
Reviewer 1 Report
Comments and Suggestions for Authors
The manuscript by Otlyga and colleagues presents a detailed histological and immunohistochemical comparison of the human adrenal medulla (AM) and organs of Zuckerkandl (OZ) during fetal development. The authors identify large and small chromaffin cell populations and trace their spatial and temporal dynamics. They provide morphological and quantitative data across gestational stages and suggest developmental and functional roles for these chromaffin structures.
General Evaluation:
This is a solid and carefully conducted histological study. The work is thorough in tissue sampling and immunohistochemical analysis and demonstrates a clear understanding of developmental anatomy. The figures are well documented, and the conclusions are largely supported by the data.
However, the manuscript falls somewhat short in terms of conceptual novelty and integration with current research directions in developmental neurobiology. While the histological observations are valuable, the lack of molecular characterization limits the broader relevance and timeliness of the study. In the era of single-cell transcriptomics and lineage tracing, more is expected than morphological classifications of “large” and “small” cells.
Major Comments:
While histological work remains fundamental, this study would benefit significantly from being placed in the context of recent transcriptomic studies (e.g., Kameneva et al., 2021; and other studies by Adameyko lab). These have begun to resolve the lineage and fate of neural crest-derived populations, including chromaffin cells. The authors mention such work only briefly, and a deeper discussion of how their findings fit, or contrast, with this emerging molecular landscape would be highly beneficial.
The classification of cells as “large” and “small” is based purely on morphology and limited immunohistochemistry. While this is historically valid, additional insights could be gained by correlating these categories with known molecular markers from single-cell datasets. Even speculative alignment would strengthen the manuscript’s conceptual impact.
Minor Comments:
Consider clarifying in the introduction why updating histological data is still important in the current molecular era.
The use of PNMT as a marker is well done, but further insights into the timing and regulation (e.g., glucocorticoid independence in humans) could be strengthened with more references and critical discussion.
Some figure legends are overly descriptive and could be condensed to highlight key points.
Comments on the Quality of English LanguageI have noticed some grammatical mistakes. For example, the first sentence in the introduction. Please, check for the grammar.
Author Response
Comment 1: While histological work remains fundamental, this study would benefit significantly from being placed in the context of recent transcriptomic studies (e.g., Kameneva et al., 2021; and other studies by Adameyko lab). These have begun to resolve the lineage and fate of neural crest-derived populations, including chromaffin cells. The authors mention such work only briefly, and a deeper discussion of how their findings fit, or contrast, with this emerging molecular landscape would be highly beneficial.
The classification of cells as “large” and “small” is based purely on morphology and limited immunohistochemistry. While this is historically valid, additional insights could be gained by correlating these categories with known molecular markers from single-cell datasets. Even speculative alignment would strengthen the manuscript’s conceptual impact.
Response 1: We sincerely appreciate the time and effort taken to evaluate our manuscript and provide valuable feedback.
We fully agree with the importance of relating our findings to recent single-cell transcriptomics data. It was very useful and interesting for us to make these parallels, and we believe this has made our work more insightful (see the Discussion, lines 459-477).
Comment 2: Consider clarifying in the introduction why updating histological data is still important in the current molecular era.
Response 2: As suggested, we have added a discussion of the continued relevance of classical histological and immunohistochemical approaches in the current molecular era to the Introduction (lines 101-117). Our results demonstrate how these traditional methods provide complementary insights to modern techniques.
Comment 3: The use of PNMT as a marker is well done, but further insights into the timing and regulation (e.g., glucocorticoid independence in humans) could be strengthened with more references and critical discussion.
Response 3: We have incorporated some critical discussion about the limitations of our PNMT-related findings. While we were unable to find direct supporting evidence in the literature, we have added some general considerations.
Comment 4: Some figure legends are overly descriptive and could be condensed to highlight key points.
Response 4: After careful consideration, we have maintained the current format of the figure legends as we believe they present the data for both morphologists and non-specialist readers. However, we would be happy to modify them if the you feel this is absolutely necessary.
Comment 5: I have noticed some grammatical mistakes. For example, the first sentence in the introduction. Please, check for the grammar.
Response 5: We have edited our English and corrected some mistakes. We hope that now its level is satisfactory.
Reviewer 2 Report
Comments and Suggestions for Authors
Review on the manuscript of Otlyga E et al., (life-3674568): “Developmental Parallels Between the Human Organs of Zuckerkandl and Adrenal Medulla”.
In this study, the authors compared the adrenal medulla and organs of Zuckerkandl during fetal development in human specimens using immunohistochemistry (IHC). The Authors identified two distinct morphological cell types (large and small cells) in the developing ganglia, organs of Zuckerkandl, and adrenal medulla. Additionally, they suggest the existence of two subpopulations of large cells, one migrating primarily from the organs of Zuckerkandl, and the other differentiating later from the smaller cells.
Overall, I find this topic to be of great interest, as the adrenal medulla and organs of Zuckerkandl play crucial roles in catecholamine production and secretion. However, the putative interaction between these two structures during human fetal development remains poorly understood. Thus, investigating the developmental processes of these organs could provide valuable information regarding their putative distinct roles and contributions to neuroendocrine function in early life, with potential consequences later in life.
I believe the Authors have addressed the primary question they proposed to explore. Below, I indicate the issues identified in the current version of the manuscript. I hope the Authors find the following comments and suggestions helpful.
1 - I kindly suggest that the Authors consider condensing the Introduction section, as it currently is very long.
2 - The organization of the Results section is challenging to follow. For example, the first part discusses Figures 1–5 and then Figure 1 is revisited in section 3.2. I would kindly suggest that the Authors consider restructuring the Results in a more sequential and coherent manner.
3 - In Figure 1C, two aortas are visible, which is unrealistic and may indicate that this section is not representative of the tissue architecture. I would kindly suggest that the Authors consider providing a representative section for better interpretation.
4 - Panels 1B, 1D, 2B, 2D, and 2F are all labeled as IHC for tyrosine hydroxylase. However, the staining patterns are different across these panels. Could the Authors kindly clarify the reason for this variation? If not due to experimental factors, I kindly suggest that the Authors provide images with consistent staining.
5 - In the Results section, the Authors state that “Immunohistochemically, LCs displayed strong cytoplasmic reactivity with antibodies for TH, DBH, and βIII-tubulin (Figure 1 A, B)”. However, the manuscript does not include images of human embryos at 8–9 g.w. stained for dopamine β-hydroxylase or βIII-tubulin. This highlights the need for a more clearly structured presentation of the figures. To facilitate comparison across developmental stages, it would be helpful to include representative IHC images for hematoxylin/eosin, tyrosine hydroxylase, dopamine β-hydroxylase, phenylethanolamine N-methyltransferase, and βIII-tubulin for embryos, pre-fetuses, and fetuses spanning from 8-9 to 25-26 g.w.. Even if some of these conditions are not essential to the core findings, they could be included as supplementary material to support a more comprehensive interpretation. Without appropriate images, it becomes difficult to compare the different conditions effectively.
6 - Another important point to consider is the number of specimens used for the interpretations presented. For example, in sections 3.2 to 3.6, are the observations and conclusions consistent across all specimens within each g.w. interval? In the Results section, the Authors state that “In one of two specimens at 12 g.w. (No. 6), intracortical chromaffin cells exhibited strong cytoplasmic positivity for PNMT (Figure 5A), while in extraadrenal chromaffin tissue and in the other specimen of similar age (No. 5), PNMT was negative”. This raises important concerns regarding the reproducibility of the findings. Given the limited number of specimens, especially for certain g.w. intervals, it is important to ensure that the results are consistent across multiple samples. Clear confirmation of reproducibility within the same gestational stages would strengthen the validity of the conclusions.
7 - In the Results section, the authors mention that “At 16 g.w. (1 specimen), no significant differences were observed compared to the 12 g.w. prefetus. However, LCs appeared negative for PNMT. This may be due to high sensitivity of PNMT to autolytic changes in tissues, which leads to instability of results”. Later, it is also mentioned “This pattern of SC location within true vessel walls or vessel-like structures may be the result of artificial changes during specimen preparation, or it could represent a true anatomical feature”. These ambiguous conclusions further highlights the importance of including a larger number of specimens to draw more robust and reliable conclusions from the data.
8 - In section 3.5, the Authors mention that “LCs were also strongly positive for PNMT, except for three of seven specimens, which appeared to be negative due to the high sensitivity of this IHC marker”. Given this variability, it becomes somewhat challenging to draw a clear conclusion from these findings. This suggests that a degree of variability already exists, which limits the strength of interpretations based on a single specimen.
9 - I kindly recommend that the Authors consider including Group 3 in the graphs shown in Figure 6 and performing the appropriate statistical analyses. This point is particularly important, as in the Discussion section the Authors state “Moreover, at the age of 22.5–26 g.w., the ratio of LCs to SCs seemed to increase”. In scientific literature, the expression “seemed to increase” is not appropriate. It would be more informative to determine whether there is a significant increase. Including Group 3 in the graphs shown in Figure 6 and performing the appropriate statistical analyses would help support a more robust and conclusive interpretation.
10 - In the Discussion section, the Authors mention that “At the age of 12 g.w., we observed strong cytoplasmic positivity for PNMT in intraadrenal LCs, both those associated with SCs and those lying freely. This finding is intriguing in light of ultrastructural results reported by Hervonen [7]”. Since in one of two specimens at 12 g.w. (No. 6), intracortical chromaffin cells exhibited strong cytoplasmic positivity for PNMT (Figure 5A), while in the other specimen of similar age (No. 5), PNMT was negative, this result is unclear. A substantial portion of the Discussion section (from line 466 to line 502) focuses on this idea. Given the limited data, and the disparity in PNMT staining between the two specimens at the age of 12 g.w., it may be helpful for the Authors to avoid this conclusion. Supporting the findings with additional data from other specimens would provide stronger evidence, as drawing firm conclusions from inconclusive results can be challenging.
Author Response
Comment 1: I kindly suggest that the Authors consider condensing the Introduction section, as it currently is very long.
Response 1: Thank you for your careful review and positive feedback on our work. We acknowledge that the introduction was indeed lengthy. This was due to our attempt to thoroughly cover all key aspects of this highly specialized research topic. Following your suggestion, we have removed the extended paragraph discussing evolutionary aspects from the Introduction, as it was not directly relevant to our study. We also deleted the second paragraph, which largely overlapped with the first.
Comment 2: The organization of the Results section is challenging to follow. For example, the first part discusses Figures 1–5 and then Figure 1 is revisited in section 3.2. I would kindly suggest that the Authors consider restructuring the Results in a more sequential and coherent manner.
Response 2: Thank you for your valuable feedback. We fully agree that the Figures and the Results may be difficult to comprehend. This can be explained by the complexity of the topic and the number of parameters involved. Having made multiple attempts to reorganize them, we came to this way of organization as the most convenient for our aims. In our results we intended to follow the method of presentation from the general to the particular. In other words, we tried to show the processes taking place in general, and then describe the details. For this reason, we were sometimes forced to return to the previously mentioned microphotographs. Other methods of reorganizing the material that we tried, unfortunately, did not allow us to display this relationship between general and local morphogenesis processes.
Below, we clarify the rationale behind the Figure organization:
- Figure 1provides an anatomical overview of the paraaortic region, highlighting key structural features. It also includes TH IHC images to illustrate the distribution of cells of interest in the adrenal primordia and OZ, as well as the TH+ cell gradient discussed later. This figure serves a dual purpose—introducing the anatomy and presenting key immunohistochemical findings—which is why it is referenced multiple times.
- Figures 2 and 3present the main IHC markers across all three age groups, facilitating comparisons of morphological and immunohistochemical features of intraadrenal LCs and SCs. Figure 2 displays H&E sections and TH IHC, while Figure 3 shows DBH and βIII-tubulin IHC. We recognize that this structure may be difficult to navigate, but it ensures comprehensive coverage of all IHC data across age groups.
- Figure 4 focuses on OZ immunostaining with the primary antibodies.
Comment 3: In Figure 1C, two aortas are visible, which is unrealistic and may indicate that this section is not representative of the tissue architecture. I would kindly suggest that the Authors consider providing a representative section for better interpretation.
Response 3: Thank you for your comment. "The effect of two aortas" is explained by the fact that due to the complex geometry of the human embryo at this stage of gestation, the aorta is curved in an arc. Due to this curve, the plane of the slice passes through two sections of the aorta above and below the origin of the renal artery, and the section of the direct origin of the renal artery remains outside the plane of the section. However, it is in this plane that the largest area of the organs we are studying (OZ and adrenal gland) is noted. That is why this particular micrograph was chosen. This bend in the aorta does not compromise the visualization of the OZ and adrenal structures.
Comment 4: Panels 1B, 1D, 2B, 2D, and 2F are all labeled as IHC for tyrosine hydroxylase. However, the staining patterns are different across these panels. Could the Authors kindly clarify the reason for this variation? If not due to experimental factors, I kindly suggest that the Authors provide images with consistent staining.
Response 4: The differences in histological staining patterns that you noticed are due to the fact that we used a shorter hematoxylin counterstain for the overview micrographs. This was done so that the counterstain would not mask the immunohistochemical reaction at such an early stage at such a magnification.
Comment 5: In the Results section, the Authors state that “Immunohistochemically, LCs displayed strong cytoplasmic reactivity with antibodies for TH, DBH, and βIII-tubulin (Figure 1 A, B)”. However, the manuscript does not include images of human embryos at 8–9 g.w. stained for dopamine β-hydroxylase or βIII-tubulin. This highlights the need for a more clearly structured presentation of the figures. To facilitate comparison across developmental stages, it would be helpful to include representative IHC images for hematoxylin/eosin, tyrosine hydroxylase, dopamine β-hydroxylase, phenylethanolamine N-methyltransferase, and βIII-tubulin for embryos, pre-fetuses, and fetuses spanning from 8-9 to 25-26 g.w.. Even if some of these conditions are not essential to the core findings, they could be included as supplementary material to support a more comprehensive interpretation. Without appropriate images, it becomes difficult to compare the different conditions effectively.
Response 5: Thank you for your comment. We organized the material into three age groups to systematically present the key IHC markers across developmental stages. In Figure 3, we focused on 11–12 gestational weeks (g.w.) for DBH and βIII-tubulin immunostaining, as this period represents the first age group in our classification. To maintain clarity, we limited the figure to one representative case per age group, as this approach effectively highlights the morphological and IHC differences between stages.
Given that TH and DBH immunostaining patterns largely overlap, we prioritized the most demonstrative micrographs—those where age-related differences are most evident. We also omitted negative results (e.g., PNMT IHC in embryos and prefetuses before 12 g.w.) to avoid redundancy.
Following your valuable suggestion, we have included additional micrographs of 8-9 gestational week (g.w.) embryos immunostained for dopamine β-hydroxylase (DBH) and βIII-tubulin in the Supplementary Materials (Supplementary Figures S1-2).
Comment 6: Another important point to consider is the number of specimens used for the interpretations presented. For example, in sections 3.2 to 3.6, are the observations and conclusions consistent across all specimens within each g.w. interval? In the Results section, the Authors state that “In one of two specimens at 12 g.w. (No. 6), intracortical chromaffin cells exhibited strong cytoplasmic positivity for PNMT (Figure 5A), while in extraadrenal chromaffin tissue and in the other specimen of similar age (No. 5), PNMT was negative”. This raises important concerns regarding the reproducibility of the findings. Given the limited number of specimens, especially for certain g.w. intervals, it is important to ensure that the results are consistent across multiple samples. Clear confirmation of reproducibility within the same gestational stages would strengthen the validity of the conclusions.
Response 6: Thank you for your insightful comment. We fully acknowledge the challenges of drawing definitive conclusions from a limited number of specimens. Unfortunately, working with human material—particularly at early gestational ages (≤12 g.w.)—presents significant difficulties due to scarcity, autolytic changes, and variability in tissue preservation.
Regarding PNMT immunostaining, while the enzyme is relatively stable compared to transcription factors, negative results in some cases may reflect material degradation or other technical limitations. However, the positive signal observed in specimen No. 6 (12 g.w.) is noteworthy, as we are unaware of any prior reports documenting the initial appearance of this enzyme in the adrenal medulla in human development. We believe this finding is unlikely to be artifactual and merits attention. Although we agree that further studies with larger cohorts are needed, we consider these preliminary results—along with our hypothesis—worthy of publication. To ensure balanced interpretation, we have included a critical discussion of these findings and their limitations in the Discussion section.
Comment 7: In the Results section, the authors mention that “At 16 g.w. (1 specimen), no significant differences were observed compared to the 12 g.w. prefetus. However, LCs appeared negative for PNMT. This may be due to high sensitivity of PNMT to autolytic changes in tissues, which leads to instability of results”. Later, it is also mentioned “This pattern of SC location within true vessel walls or vessel-like structures may be the result of artificial changes during specimen preparation, or it could represent a true anatomical feature”. These ambiguous conclusions further highlights the importance of including a larger number of specimens to draw more robust and reliable conclusions from the data.
Response 7: Thank you for your comment. Regarding PNMT, please refer to our response to comment No. 6.
Concerning the observed pattern of (SC) localization, our findings are supported by existing literature (references [21, 22]). We propose that this phenomenon likely represents a consistent tissue artifact at specific developmental stages. This interpretation is further supported by ultrastructural studies in various species, including humans, where similar features have not been observed at the electron microscopic level.
Comment 8: In section 3.5, the Authors mention that “LCs were also strongly positive for PNMT, except for three of seven specimens, which appeared to be negative due to the high sensitivity of this IHC marker”. Given this variability, it becomes somewhat challenging to draw a clear conclusion from these findings. This suggests that a degree of variability already exists, which limits the strength of interpretations based on a single specimen.
Response 8: Thank you for your observation. We acknowledge the presence of some variability in our findings, which indeed warrants further investigation. Regarding the specific aspects of PNMT expression, please refer to our detailed response to comment No. 6 for additional clarification.
Comment 9: I kindly recommend that the Authors consider including Group 3 in the graphs shown in Figure 6 and performing the appropriate statistical analyses. This point is particularly important, as in the Discussion section the Authors state “Moreover, at the age of 22.5–26 g.w., the ratio of LCs to SCs seemed to increase”. In scientific literature, the expression “seemed to increase” is not appropriate. It would be more informative to determine whether there is a significant increase. Including Group 3 in the graphs shown in Figure 6 and performing the appropriate statistical analyses would help support a more robust and conclusive interpretation.
Response 9: Thank you very much for your comment. Yes, indeed, our conclusion sounds rather incorrect. Given the very small number of samples in the third group (2 cases), it was impossible to perform a relevant statistical analysis. Therefore, plotting graphs showing the change in cell counts would likely be misleading. To address this, we have revised our conclusion to avoid overinterpretation. Instead of quantitative claims, we now provide only a qualitative morphological description, using the term "visually increased" to reflect our observations. Additionally, we explicitly emphasize in the text that further studies on a larger number of samples are required to confirm this morphological trend.
Comment 10: In the Discussion section, the Authors mention that “At the age of 12 g.w., we observed strong cytoplasmic positivity for PNMT in intraadrenal LCs, both those associated with SCs and those lying freely. This finding is intriguing in light of ultrastructural results reported by Hervonen [7]”. Since in one of two specimens at 12 g.w. (No. 6), intracortical chromaffin cells exhibited strong cytoplasmic positivity for PNMT (Figure 5A), while in the other specimen of similar age (No. 5), PNMT was negative, this result is unclear. A substantial portion of the Discussion section (from line 466 to line 502) focuses on this idea. Given the limited data, and the disparity in PNMT staining between the two specimens at the age of 12 g.w., it may be helpful for the Authors to avoid this conclusion. Supporting the findings with additional data from other specimens would provide stronger evidence, as drawing firm conclusions from inconclusive results can be challenging.
Response 10: Thank you for your valuable comment. As addressed in our response to comment No. 6, we believe retaining the discussion of our hypothesis is important for the scientific discourse. While we fully acknowledge the need for additional specimens to strengthen these findings, we have found no evidence to question the validity of the positive result. Nevertheless, we have incorporated a thorough critical discussion of our hypothesis to provide proper context and balance to our interpretation.
Round 2
Reviewer 2 Report
Comments and Suggestions for Authors
Second review on the manuscript of Otlyga E et al., (life-3674568): “Developmental Parallels Between the Human Organs of Zuckerkandl and Adrenal Medulla”.
In this study, the authors compared the adrenal medulla and organs of Zuckerkandl during fetal development in human specimens using immunohistochemistry (IHC). The Authors identified two distinct morphological cell types (large and small cells) in the developing ganglia, organs of Zuckerkandl, and adrenal medulla. Additionally, they suggest the existence of two subpopulations of large cells, one migrating primarily from the organs of Zuckerkandl, and the other differentiating later from the smaller cells.
This is the second version of the manuscript following peer review. After a careful review of the revised manuscript, I acknowledge the Authors for the clarifications provided and the improvements made to the manuscript. However, there are still some points that could benefit of an improvement. Below, I outline the issues identified in the current version of the manuscript. I hope the Authors find the following comments and suggestions helpful.
1 - While I understand that Group 3 includes only two samples, which limits the possibility of statistical analysis, it could still be helpful to include this data in the graphs in Figure 6 to give readers a visual impression of the potential effect.
2 - In the figures, the scale bars differ between panels. I kindly recommend using a consistent scale bar size across all panels. 100 µm could be an appropriate choice for uniformity.